# Plasticity of Response Properties of Mouse Visual Cortex Neurons Induced by Optogenetic Tetanization In Vivo

Ivan V. Smirnov [1],[†] , Aksiniya A. Osipova [1],[†], Maria P. Smirnova [1] , Anastasia A. Borodinova [1] ,
Maxim A. Volgushev [2] and Alexey Y. Malyshev [1],*

[1]  Institute of Higher Nervous Activity and Neurophysiology of RAS, Moscow 117485, Russia;
ivan.vas.smirnov@gmail.com (I.V.S.); aksiniyaosipova@gmail.com (A.A.O.); rymarik@gmail.com (M.P.S.);
borodinova.msu@mail.ru (A.A.B.)
[2]  Department of Psychological Sciences, University of Connecticut, Storrs, CT 06269, USA;
maxim.volgushev@uconn.edu
*  Correspondence: malyshev@ihna.ru
[†]  These authors contributed equally to this work.

**Abstract:** Heterosynaptic plasticity, along with Hebbian homosynaptic plasticity, is an important mechanism ensuring the stable operation of learning neuronal networks. However, whether heterosynaptic plasticity occurs in the whole brain in vivo, and what role(s) in brain function in vivo it could play, remains unclear. Here, we used an optogenetics approach to apply a model of intracellular tetanization, which was established and employed to study heterosynaptic plasticity in brain slices, to study the plasticity of response properties of neurons in the mouse visual cortex in vivo. We show that optogenetically evoked high-frequency bursts of action potentials (optogenetic tetanization) in the principal neurons of the visual cortex induce long-term changes in the responses to visual stimuli. Optogenetic tetanization had distinct effects on responses to different stimuli, as follows: responses to optimal and orthogonal orientations decreased, responses to null direction did not change, and responses to oblique orientations increased. As a result, direction selectivity of the neurons decreased and orientation tuning became broader. Since optogenetic tetanization was a postsynaptic protocol, applied in the absence of sensory stimulation, and, thus, without association of presynaptic activity with bursts of action potentials, the observed changes were mediated by mechanisms of heterosynaptic plasticity. We conclude that heterosynaptic plasticity can be induced in vivo and propose that it may play important homeostatic roles in operation of neural networks by helping to prevent runaway dynamics of responses to visual stimuli and to keep the tuning of neuronal responses within the range optimized for the encoding of multiple features in population activity.

**Keywords:** neuron; heterosynaptic plasticity; visual cortex; optogenetics; mice; in vivo

## 1. Introduction

Synaptic plasticity represents the cellular basis of learning. Synaptic plasticity is not a uniform phenomenon; rather, multiple forms and mechanisms have been described and studied [1,2]. Synaptic plasticity can be segregated into two broad groups, homosynaptic and heterosynaptic, defined by the requirement of presynaptic activity at a synapse during the induction. Homosynaptic plasticity requires presynaptic activity at the synapse during the induction and, thus, occurs at synapses that were directly involved in the activation of a cell during the induction. Homosynaptic plasticity can be associative, such as canonical NMDA-dependent plasticity in the CA1 area of the hippocampus [3], or non-associative, such as plasticity at mossy fiber-CA3 synapses [4]. Heterosynaptic plasticity does not require presynaptic activation of the synapse during the induction and, thus, can occur at synapses that were not active during the induction [5]. In this paper, we are following this use of the terms homosynaptic and heterosynaptic, which is conventional for the research of plasticity in mammalian nervous systems [6]. Relation to the terminology

used in invertebrate research [7–9] will be considered in the discussion. In short, the term heterosynaptic plasticity in invertebrate research embraces a broader range of phenomena, including associative changes, non-associative changes, and modulatory plasticity.

Distinct forms of plasticity play diverse functional and computational roles. Homosynaptic plasticity can be Hebbian and associative and can, thus, mediate associative learning. Its mechanisms are studied in great detail, in diverse experimental preparations, including the plasticity of the response properties and receptive fields of neurons in the visual cortex in vivo [10–12]. However, Hebbian-type rules introduce positive feedback on changes of synaptic weights and activity. Indeed, theoretical studies and computer simulations have shown that model neural networks built using only Hebbian-type plastic synapses are intrinsically unstable, with synaptic weights tending to reach extreme saturated values and activity prone to runaway dynamics, either an unrestrained increase or silencing. To counteract these undesired effects, theoretical and simulation studies suggested heterosynaptic plasticity as a necessary component of learning networks. Indeed, incorporating heterosynaptic plasticity in the models helps to robustly prevent runaway dynamics, enhance synaptic competition, and increase the contrast of synaptic weight changes [5,13–15]. Heterosynaptic plasticity has received less attention than homosynaptic plasticity and has been predominantly studied in slice preparations, which allow for better control of stimulation, especially for the absence of presynaptic activity during the induction. In slices from the hippocampus, heterosynaptic long-term depression (LTD) accompanying tetanus-induced long-term potentiation (LTP), but occurring at non-activated synapses was first described by Lynch and colleagues [6]. A feasible trigger for this form of heterosynaptic plasticity is bursts of action potentials [16,17], which are generated during the induction of homosynaptic LTP and back-propagate into the dendritic tree [16]. This type of heterosynaptic plasticity requires the activation of L-type voltage-dependent calcium channels but does not depend on the activation of glutamate NMDA receptors [16], which act as molecular detectors of the coincidence of both pre- and postsynaptic activity and are critical for canonical homosynaptic Hebbian plasticity [18]. On a local scale, heterosynaptic plasticity could be induced by the diffusion of diverse signaling molecules within the dendrite or in the extracellular space, causing a Mexican hat-shaped profile of LTP and LTD around the location of the tetanized synapses [19–21], a breakdown of input specificity of plastic changes [22], and even a spread of plasticity to closely located but non-stimulated neurons after pairing [23]. An established experimental paradigm to study heterosynaptic plasticity is intracellular tetanization—bursts of spikes induced by depolarization pulses without presynaptic stimulation. Plastic changes induced by such a purely postsynaptic protocol can be interpreted as heterosynaptic. The intracellular tetanization paradigm has been applied to study the plasticity in diverse cells and preparations, including pyramidal neurons in the hippocampus [17], granular neurons of the dentate gyrus [24], excitatory and inhibitory neurons of the neocortex [25–27], and even identified neurons of the common snail [28], but has never been tested in in vivo preparations. In contrast to in vitro preparations, evidence for heterosynaptic plasticity in vivo is sparse, in part because of difficulties in controlling for the absence of presynaptic stimulation during plasticity induction. In the visual cortex, heterosynaptic changes were described in experiments using two-photon microscopy imaging of calcium signals in spines. In the mouse visual cortex, induction of homosynaptic LTP, by pairing the activation of spines using visual stimuli with depolarization-evoked postsynaptic spikes, was accompanied by heterosynaptic LTD in some other spines at the same neuron [29]. While the absence of presynaptic activation during the induction was not controlled for and cannot be excluded, especially considering the huge size of the receptive fields studied and the strong visual stimulation used, these data indicate that heterosynaptic plasticity can be induced in the mouse visual cortex in vivo.

Here, we set out to resolve the question of whether heterosynaptic plasticity can be induced in vivo. A drawback of using visual or electric stimulation to evoke the strong postsynaptic firing necessary for plasticity induction in visual cortex neurons is that such

stimuli are associated with the activation of a large set of presynaptic fibers that may include the test inputs. This would hinder the clear interpretation of plastic changes at the test inputs as heterosynaptic. To circumvent this limitation, we used a postsynaptic induction protocol of intracellular tetanization—bursts of postsynaptic spikes evoked using optogenetic stimulation of the recorded neuron without presynaptic activation. Thus, for the first time, we used the protocol of intracellular tetanization, which is well-established in vitro, to study the possible role of heterosynaptic plasticity in the functioning of neuronal networks in vivo. To achieve lasting recordings and to avoid another potential problem of 'plasticity wash-out' we employed juxtacellular recording. We show that, in the principal neurons of the mouse visual cortex, bursts of high-frequency action potentials without presynaptic activation can induce long-term changes in the responses to visual stimuli and a change in the directional preference of neurons. Because there was no visual stimulation and presynaptic activation during the induction, the observed changes in the responses were mediated by heterosynaptic plasticity. We conclude that heterosynaptic plasticity can be induced in vivo and propose that it could play a vital role in maintaining the stability of operation and functionality of learning networks in vivo.

## 2. Materials and Methods

### 2.1. Animals

Experiments were performed using adult C57Black/6 mice, 2–4 months of age (Pushchino Breeding Center, Branch of the Shemyakin–Ovchinnikov Institute of bioorganic chemistry of RAS). All experimental procedures were in accordance with the 2010/63/EU directive for the use of laboratory animals. The research protocol was approved by the Ethics Committee of the Institute of Higher Nervous Activity and Neurophysiology, Russian Academy of Sciences (approved on 15 January 2020; protocol No. 1; project # 20-15-00398).

### 2.2. AAV Virus Production and Injection

The virus injection was made under isoflurane anesthesia. Dexamethasone (2 mg/kg, intracutaneous) was used to prevent inflammation after the surgery. The recombinant adeno-associated viruses serotype 2 (AAV2) for the expression of channelrhodopsin oChIEF fused with EGFP reporter protein under control of the CaMKIIa promoter were produced at local facilities, according to a standard protocol. The virus was slowly injected intracranially in the right hemisphere at stereotaxic coordinates (A/P = −2.8, M/L = 1.4 from Bregma), 200 μm below the brain surface (1 μL in concentration of $1.49 \times 10^{12}$ vg/mL in PBS, injection rate 0.06 μL per minute) (Figure 1A).

### 2.3. Surgery

Recording experiments were made 2–4 weeks after viral injection. During surgery and recording, the animal was kept under urethane anesthesia (0.5–1.5 g/kg). Additionally, atropine (5 mg/kg) and dexamethasone (2 mg/kg) were administered subcutaneously to reduce secretion and edema. During the surgery, the head of the animal was fixed in the head holder SMG-4 (Narishige, Tokyo, Japan). To improve the signal-to-noise ratio during intrinsic imaging, the scull over V1 was thinned out. The head plate was affixed to the skull with dental cement on cyanoacrylate glue. After one hour of recovery from surgery, the mouse was placed in the experimental setup.

### 2.4. Visual Stimulation

Visual stimuli were presented on an LED monitor (mean luminance 45 Cd, 60 Hz), placed 25 cm from the mouse, covering ~70° of visual space. Visual stimuli, drifting sine-wave gratings (0.04 cpd, 2 Hz, 12 directions), were generated using PsychoPy [30]. During electrophysiological recording, drifting gratings of 12 orientations were presented in a pseudorandom order, each grating presented for 2 s with 0.5 s interval (gray screen) between the stimuli. Sets of stimuli moving in all 12 directions were repeatedly presented during the recording.

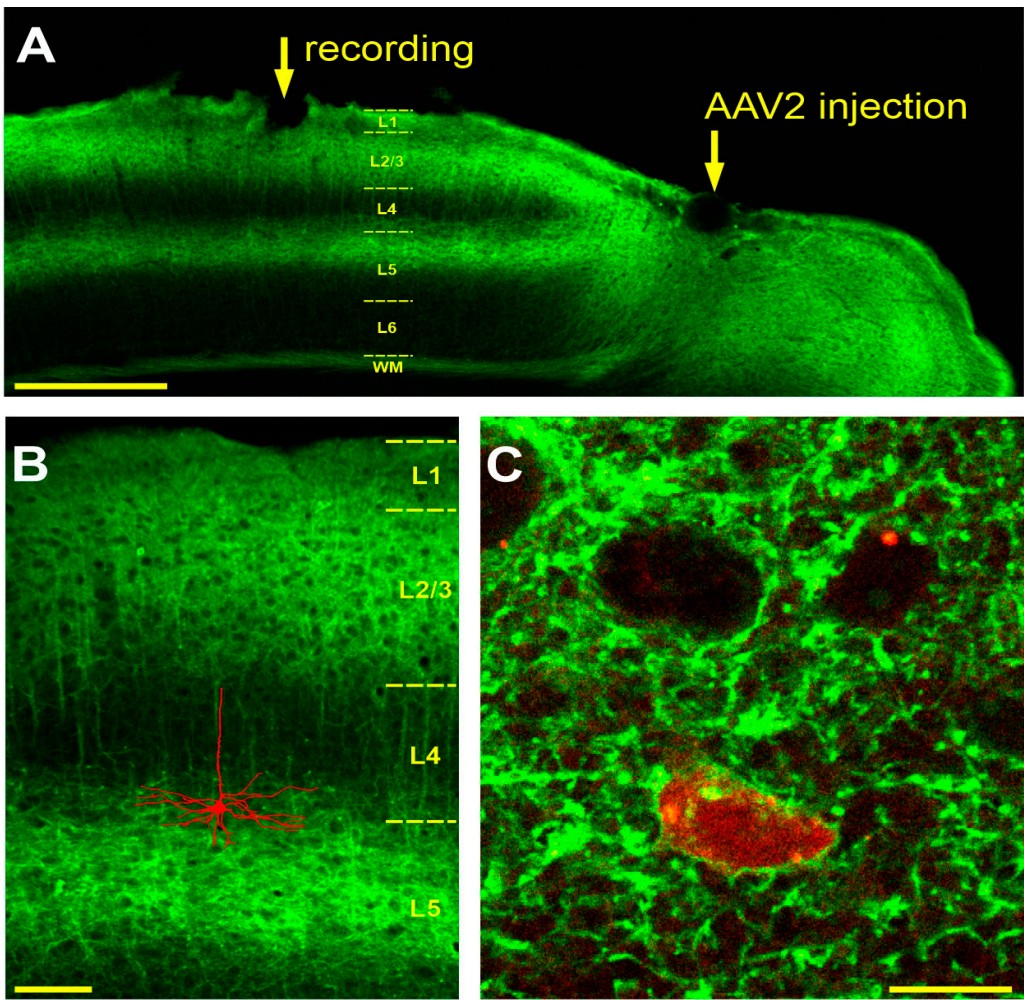

**Figure 1.** Expression of oChIEF-EGFP in mouse visual cortex. (**A**)—Frontal section of the mouse visual cortex after transduction with the AAV2-CamKII-oChIEF-EGFP virus, with the sites of virus injection and electrophysiological recording (2 weeks after the injection) indicated by the arrows. Note: strong fluorescence in all cortical layers around the recording site is observed, but with increasing distance from the injection site only in layers 1, 2/3, the upper portion of layer 5, and in the white matter (WM). The boundaries between cortical layers (L1, L2/3, L4, L5, L6) and white matter (WM) are shown schematically. (**B**)—Pyramidal morphology of a neuron stained with neurobiotin during recording and then visualized with Streptavidin-Rhodamin. The image was superimposed with an image of a neighboring slice from the same animal showing the oChIEF expression pattern. Both images were taken using a confocal microscope, automatically traced and scaled. (**C**)—Soma of a neuron stained during recording with neurobiotin and visualized with Streptavidin-Rhodamine. A single confocal section. Green fluorescence of the somatic membrane is indicative of oChIEF-EGFP expression. Calibration bars: (**A**)—500 μm, (**B**)—100 μm, (**C**)—10 μm.

### 2.5. Intrinsic Optical Imaging

Prior to electrophysiological recording, the retinotopic projection of the monitor onto V1 was determined using intrinsic optical signal (IOS). The IOS was recorded using a CMOS camera (BFS-U3-17S7M-C, FLIR, Wilsonville, OR, USA; 40 fps) and Micro-Management software. During the first 2 s of the trial, background frames were collected while a grey screen was presented, then visual stimuli (drifting gratings, 0.5 s for each of 12 orientations) were presented for 6 s and then a grey screen was presented for 12 s. A total of 30 of such 20 s trials were collected. Each frame was then normalized to mean intensity and the average background frame was subtracted from the averaged response frame. Using the resulting image, the center of the retinotopic projection of the screen was determined as the area

with the lowest intensity. A small craniotomy (300–500 μm) was then made in the center of the area corresponding to the retinotopic projection of the screen.

*2.6. Electrophysiological Recording*

Recordings were started at least 2 weeks after the virus injection. For electrophysiological recordings, we used a juxtacellular technique. Borosilicate glass electrodes were filled with Hanks' Balanced Salt Solution (HBSS, 138 mM NaCl, 1.26 mM $CaCl_2$, 0.5 mM $MgCl_2$, 0.4 mM $MgSO_4$, 5.3 mM KCl, 0.44 mM $KH_2PO_4$, 4.16 mM $NaHCO_3$, 0.34 mM $Na_2HPO_4$, 10 mM glucose, and 10 mM HEPES at pH 7.4) and had a resistance of 4–12 MΩ. An optic fiber (200 μm core) was inserted in the recording electrode using a specialized electrode holder (Optopatcher, A-M Systems, Sequim, WA, USA) and connected to a 470 nm LED (ThorLabs, Newton, NJ, USA). The electrode with the optic fiber was inserted into the brain through a small opening in the dura (about 100 μm), made in the middle of the craniotomy.

Light intensity, measured at the tip of the optical fiber with the maximal LED power, was around 5 mW (power meter PM16-130, ThorLabs). Only the very tip of the recording pipette, about 500 μm or less, was inserted into the brain through a small opening in the dura. Since the optical fiber did not reach the tip of the glass pipette, being shorter by about 2.5 mm, most of the light emitted from the tip of the fiber was falling onto the surface of the skull and the dura and only a small portion of the beam entered the brain (Figure 2A inset). To estimate the light power that would reach the brain in such a configuration, we made a small hole (about 100 μm) in a piece of paper, lowered the electrode 500 μm into this hole and measured the light intensity using the power meter. These measurements showed that the total light power that reached the brain in our recording/optostimulation configuration was approximately 130 μW.

During the cell search, short light pulses were constantly delivered through the optic fiber. Only neurons that generated spikes in response to every optogenetic stimulus (i.e., oChIEF-expressing neurons) were taken in the experiment. The experiment was performed as follows: First, control responses to moving gratings were recorded for 15–40 min. Then, visual stimulation was stopped and the recorded neuron was either optogenetically tetanized (tetanus group) or was recorded without any stimulation for 5–6 min (control group). Control experiments were performed on the non-transduced mice. Optogenetic tetanization consisted of 5 series of 10 trains of light pulses delivered through the optic fiber. The inter-series interval was 1 min and the inter-train interval was 1 s. Each train consisted of 8–10 pulses with the frequency of 75–100 Hz and pulse duration of 8 ms. The frequency and the number of pulses were adjusted individually for each cell, in order to induce APs with the target frequency of 100 Hz or the maximal possible frequency for that cell. Since the level of oChIEF expression varies between cells, not all recorded neurons were able to follow the target spiking frequency of 100 Hz. In such cases, we lowered the frequency of the light stimuli during optogenetic tetanization to the maximal which the cell could follow, but not lower than 75 Hz. During tetanization, no visual stimuli were presented. In the control group, there was a pause in the visual stimulation for 5–6 min, corresponding to the duration of optogenetic tetanization. After the tetanization (or a pause in the stimulation in the control group), visual stimulation was resumed and responses to moving gratings were recorded for at least 1 h. Recordings were made using a Multiclamp 700B amplifier (Molecular Devices, San Jose, CA, USA) with PClamp 10 software (Molecular Devices, USA) and the signals were filtered at 300–10,000 Hz and were digitized at 20 kHz (Digidata 1550 Series, Molecular Devices, USA).

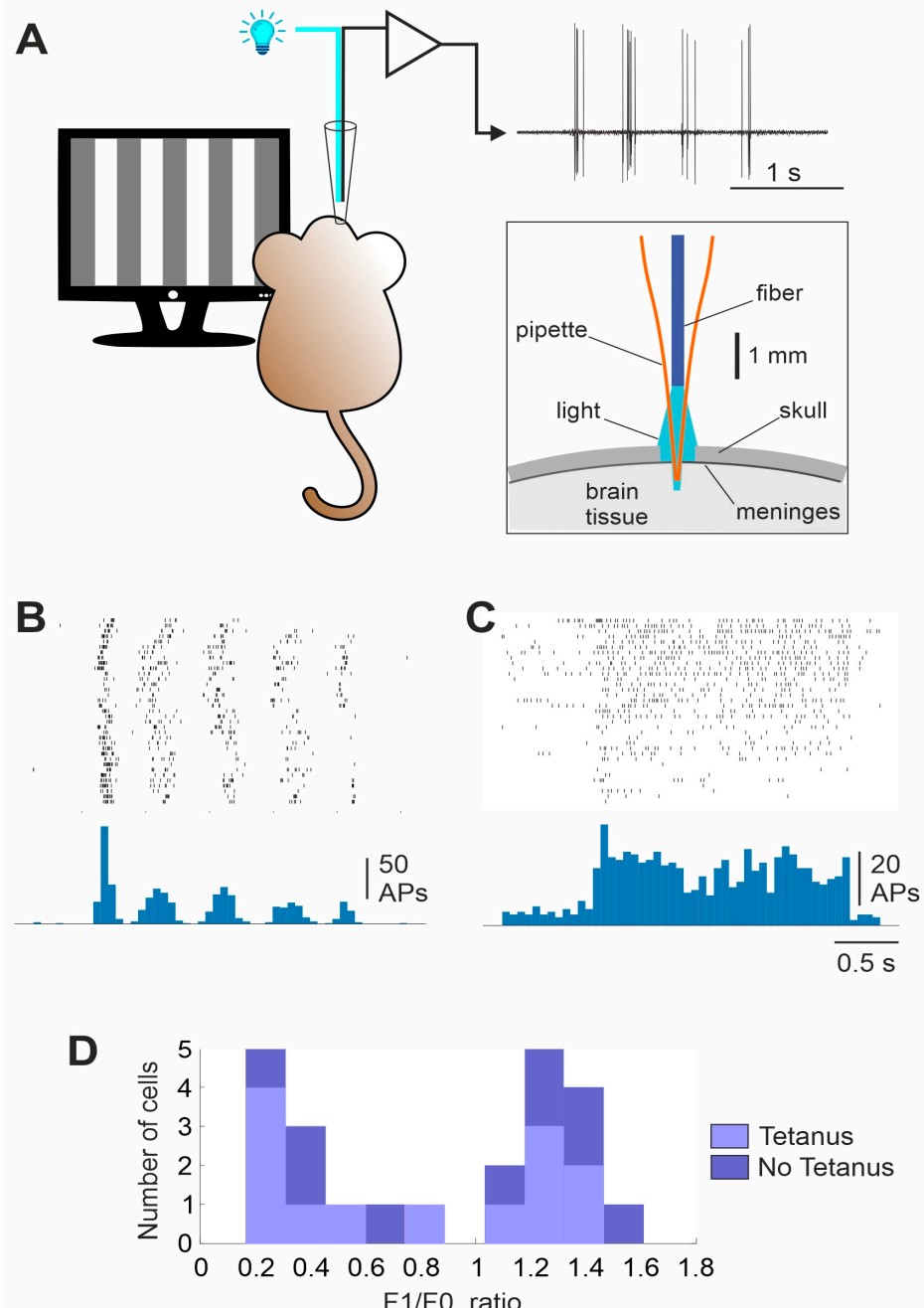

**Figure 2.** Responses of neurons with simple and complex receptive fields to visual stimulation. (**A**)—A scheme of the experiment. For local optogenetic stimulation, an optical fiber was inserted into the recording glass electrode alongside a silver wire for the recording of neuronal activity. Inset: schematic illustrating the propagation of light from the Optopatcher during optogenetic tetanization. Only a small fraction of the light emitted from the fiber reaches the brain. (**B,C**)—Examples of responses of a simple (**B**) and a complex (**C**) cell to a moving grating of optimal orientation. Raster diagrams of spike responses to 35 presentations of the moving grating and peri-stimulus time histograms plotted using the spikes from the rosters. Note the pronounced modulation of the response at the stimulus frequency (2 Hz) in a simple cell (**B**), but little modulation in a complex cell (**C**). (**D**)—Distribution of the ratio of the modulation component of the response to the average response amplitude (F1/F0 ratio) for cells in which optogenetic tetanization was applied (Tetanus group) and cells in the control (No Tetanus) group. A clear bimodal distribution allows for a clear formal classification of cells as simple (F1/F0 > 1) or complex (F1/F0 < 1). Note: an approximately equal number of simple and complex cells in the tetanus and no tetanus groups.

### 2.7. Labeling

To label recorded neurons, 1% neurobiotin was added to the electrode solution in some experiments (Vector Laboratories, Newark, CA, USA). To deliver neurobiotin into the neurons, 500 ms positive current pulses at 2 Hz frequency were applied during the last 10 min of the recording. The amplitude of the current pulses was 2–10 nA, adjusted individually for each neuron. After allowing neurobiotin to diffuse in the cell for one hour, mice were perfused with 10% PFA, then the brain was extracted and post-fixed in 4% PFA overnight. 50 µm slices were made using a vibratome (Leica 1100, Leica, Wetzlar, Germany). Neurobiotin was visualized by incubating slices in 1:250 Streptavidin-Rhodamin (ThermoFisher, Waltham, MA, USA) solution in PBS. The slices were examined using a laser confocal Cerna-based microscope (Thorlabs, USA).

### 2.8. Data Analysis

Data analysis was carried out using custom scripts in Python3. APs were detected using a threshold which was set individually for each cell. To calculate the F1/F0 ratio, the peristimulus time histogram of visual responses was fitted with a 2 Hz sine wave (corresponding to the spatial frequency of the moving gratings) and then the amplitude of the fitted sine wave was divided by the average amplitude of the response. The direction and orientation selectivity indices (DSI and OSI) were calculated as follows:

$$DSI = (Ropt - Rnull)/(Ropt + Rnull), \quad OSI = (Ropt - Rort)/(Ropt + Rort)$$

where Ropt is the response to the visual stimulus of the optimal orientation and direction; Rnull is the response to the visual stimulus of the optimal orientation but moving in the opposite direction to the optimal one; and Rort is the averaged response to the visual stimuli, orthogonal to the optimal orientation, moving in both directions. For the experimental and control groups in which distribution of the data did not pass the normality test, we used nonparametric tests (e.g., the Wilcoxon test), and for the data which passed normality tests, parametric tests (e.g., the *t*-test) were used. Specific tests are indicated in the results. Throughout the paper, values are presented as mean ± SD.

### 3. Results

The aim of our work was to investigate whether a postsynaptic challenge—bursts of action potentials without presynaptic stimulation (unpaired tetanization)—can induce changes in the response properties of neurons in the mouse visual cortex in vivo. To achieve stable long-term recordings from individual neurons, without compromising the integrity of their membrane, while having a possibility to stimulate them through the recording pipette, we first used the technique of juxtacellular recording combined with optogenetic stimulation. However, a pilot series of experiments showed that passing through a juxtacellular electrode current pulses of high amplitude, necessary for the induction of high-frequency bursts of action potentials, resulted in membrane rupture. To circumvent this problem, we decided to use an optogenetic approach for the tetanization of neurons. Classical channelrhodopsin2, due to the pronounced sensitization of responses during rhythmic stimulation, allows for the induction of controlled bursts of action potentials at frequencies up to 20–30 Hz, but not higher [31]. Therefore, we used the fast channelrhodopsin oChIEF, which allows the efficient optogenetic stimulation of neurons at frequencies around 100 Hz [32], thus allowing us to reproduce conventional plasticity-induction protocols. Previously, it has been shown that high-frequency light stimulation can reliably induce classical LTP in oChIEF-expressing neurons [32]. To transduce neurons in the mouse primary visual cortex with fast channelrhodopsin oChIEF, we injected the AAV2-CaMKII-oChIEF-EGFP virus medio-caudal to the intended recording site, at least 2 weeks before the recording. The morphological control made after electrophysiological experiments revealed strong fluorescence in all cortical layers around the site of injection (Figure 1A). At the recording sites, 1–2 mm away from the injection site in the lateral–posterior direction, the fluorescence pattern was not uniform across the layers. There was a pronounced fluorescence in layers

1, 2/3, and 5a of the visual cortex and in the white matter (Figure 1A). In layer 4, the fluorescence was very weak, only in the dendrites of the layer 5 pyramids (Figure 1A,B). In layer 6 and in the lower portion of layer 5 (5b), fluorescence was absent.

In four experiments, neurons expressing oChIEF, as verified by the ability of light pulses to evoke action potentials, were labeled for morphological identification. In such experiments, neurobiotin was added to the electrode solution and electroporated into the cell at the end of the recording. Three cells were successfully stained and demonstrated a typical morphology of layer 5 pyramidal neurons (Figure 1B). A morphological analysis carried out using confocal microscopy showed that all cells stained during the recording with neurobiotin and visualized with Rhodamine showed green fluorescence in the near-membrane region (Figure 1C). Thus, expression of the oChIEF-EGFP construct in these cells is verified using both physiological and morphological approaches.

At the beginning of each experiment, we recorded neurons' control responses to sine-wave gratings moving in 12 different directions (Figure 2A). We observed two types of responses to the moving gratings. Responses of the first type were characterized by a strong modulation of the visually induced activity at the temporal frequency of the stimulus (Figure 2B). Responses of the second type had a predominantly tonic increase in the spiking frequency during the stimulus presentation, but little modulation at the stimulation frequency (Figure 2C). These distinct patterns of responses are characteristic features of visual cortex neurons with simple and complex receptive fields, well-described in different mammalian species and used for the classification of cortical receptive fields [33–36]. For the formal classification of neurons, we calculated the F1/F0 ratio using the response to the optimal orientation. The envelope of the peri-stimulus time histogram was fitted using a sinusoid with the frequency of the stimulus. The F1/F0 ratio was calculated as the ratio of the amplitude of the fitted sine wave to the average amplitude of the response. The F1/F0 ratio of the recorded neurons showed a bimodal distribution, allowing a clear classification of neurons into simple (F1/F0 > 1, $n = 11$) and complex (F1/F0 < 1, $n = 12$) cells (Figure 2D). The proportion of simple and complex cells in the experimental and control groups (see below) did not differ significantly (seven simple and six complex cells in the experimental group and four simple and six complex cells in the control group; Chi-square = 0.4343, $p = 0.51$, $n = 10$ animals in each group).

Orientation tuning of a neuron was calculated using averaged responses (number of spikes) to stimuli of different orientations. To facilitate comparison across cells, orientation tuning of each neuron was normalized by the averaged response to all orientations of the moving gratings. Examples of typical orientation tunings of two neurons, plotted in polar coordinates, are shown in Figure 3A. After recording responses to moving gratings for 15–40 min, visual stimulation was stopped and either an optogenetic tetanization of the neuron was performed (experimental group), or no action was taken (control group). The optogenetic tetanization consisted of 5 series of 10 trains of 8–10 pulses (8 ms, 75–100 Hz) of 470 nm light, applied through the optical fiber inserted into the recording electrode. Each train of the optical tetanization evoked in the recorded neurons bursts of action potentials with a frequency of 75–100 Hz (Figure 3B). Optogenetic tetanization lasted 5–6 min. After the optogenetic tetanization, or after 5–6 min without any stimulation in the control group, visual stimulation with moving gratings of different orientations was resumed and continued for at least one hour.

In the analysis, we first asked whether the optimal stimulus (grating of the optimal orientation moving in the preferred direction) changes after tetanization. Figure 3C shows changes of the optimal orientation of the moving grating after tetanization and in the control group, plotted against the F1/F0 ratio. Note that because 30 deg was the step of orientations used during testing, changes by 30 deg could have been spurious, since the 'true' optimal orientation could be between the tested ones. With this consideration in mind, the most frequent change was that the optimal (strongest) response after tetanization was evoked by stimuli of the same orientation but moving in the opposite (previously null) direction (Figure 3C).

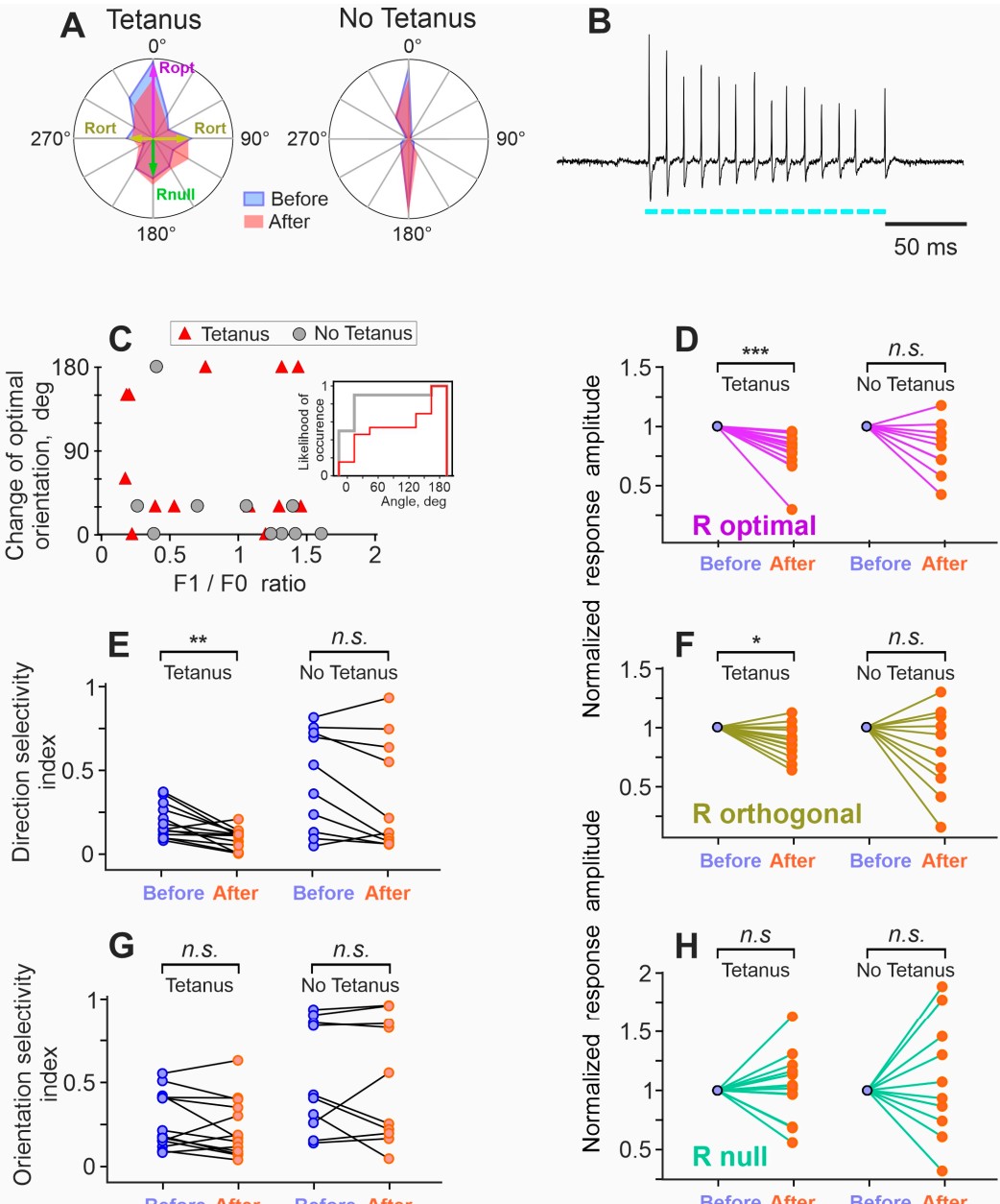

**Figure 3.** Optogenetic tetanization induces changes in the responses to visual stimulation and functional properties of neurons in the visual cortex. (**A**)—Examples of orientation tuning of a neuron subject to optogenetic tetanization (Tetanus group) and a neuron from the control (No Tetanus) group. For both neurons, tunings before (blue) and after (pink) the tetanization, or a respective time interval without stimulation for the cell from the control group, are shown. For the neuron in the Tetanus group, optimal (Ropt), orthogonal (Rort), and null orientations (Rnull) are indicated. (**B**)—A burst of action potentials induced by optogenetic tetanization in a neuron expressing oChIEF. The cyan bars below the trace show the timing and duration of the light stimuli delivered through the optical fiber inserted in the glass electrode. (**C**)—Changes of the optimal orientation after tetanization (or stimulation pause) plotted against the F1/F0 ratio. Note that changes of the optimal orientation occurred much more frequently after the tetanization (Tetanus group) than in the control (No Tetanus) group. Note also that the change in the optimal orientation did not depend on the F1/F0 ratio. Inset: cumulative distributions of the optimal orientation change occurrence in the Tetanus (red) and No Tetanus (gray) groups. (**D,F,H**)—Changes of the normalized amplitude of responses to the stimuli of optimal (**D**), orthogonal (**F**), and null (**H**) orientation after tetanization (Tetanus, left parts of the plots)

and in the control (No Tetanus, right parts of the plots) group. Response amplitude before tetanization (or before a pause in stimulation) was taken as one; each pair of points connected by a line represents data from one cell. Color code as in (**A**). * $p < 0.05$, ** $p < 0.01$, *** $p < 0.001$, n.s.—not significant. (**E,G**)— Changes in the indices of direction selectivity (**E**) and orientation selectivity (**G**) of visual cortex neurons after optogenetic tetanization (Tetanus) and in the control group (No Tetanus). For those cells, in which the preferred orientation changed after the tetanization (or a pause in stimulation), the new, re-defined 'optimal' direction was used for the calculation of parameters of the 'After' responses.

In the tetanus group, the change in the optimal orientation of the stimulus occurred more frequently than in the control (no tetanus) group. There was a significant difference between distributions of the optimal orientation changes in the tetanus and no tetanus groups ($p < 0.05$, Anderson–Darling test; Figure 3C inset). The change in the orientation of the optimal stimulus after tetanization did not correlate with the F1/F0 index (Figure 3C) and, thus, occurred in both simple and complex cells. As described above, we classified cells with F1/F0 > 1 as simple, and with F1/F0 < 1 as complex (Figure 2D). In the tetanization group, the F1/F0 in simple cells was, on average, $1.3 \pm 0.14$ ($n = 6$), while in complex cells, it was $0.35 \pm 0.22$ ($n = 7$).

Next, we compared responses to each orientation of the grating before and after the tetanization (or after 5–6 min without stimulation in the control group). Optogenetic tetanization had differential effects on the responses to different orientations.

After tetanization, the amplitudes of responses to the gratings of optimal orientation significantly decreased to $77.7 \pm 17.3\%$, compared to the values before tetanization ($p < 0.001$, effect size r $= -1.017$, $n = 13$, Wilcoxon matched-pairs signed rank test; Figure 3D, left panel). Responses to the non-optimal (orthogonal to optimal) orientation decreased to $87.8 \pm 13.9\%$ ($p < 0.0105$, r $= -0.71$, Figure 3F, left panel). In the control (no tetanus) group, responses to neither the optimal orientation, nor to the non-optimal orientation changed significantly ($84.7 \pm 25\%$, $p = 0.13$, r $= -0.48$., and $89.1 \pm 27.8\%$, $p = 0.43$, r $= -0.25$; $n = 10$; Figure 3D,F, right panels). Responses to the null direction of movement (grating of optimal orientation but moving in the opposite to preferred direction) did not change significantly in the two groups ($113 \pm 35\%$, $p = 0.17$, r $= 0.38$ for the experimental group, $111 \pm 47.6\%$, $p = 0.62$, r $= 0.15$ for the control group, Figure 3H). As a consequence of the decrease in responses to the optimal stimulus, but unchanged response to the stimulus moving in the null direction, there was a significant decrease in the directional selectivity index after optogenetic tetanization, which decreased to $51.5 \pm 38.2\%$ of the values before tetanization, from $0.22 \pm 0.15$ to $0.096 \pm 0.06$, ($p < 0.0017$, r $= -0.87$, Wilcoxon matched-pairs signed rank test). In the control (no tetanus) group, the direction selectivity index did not change ($p = 0.11$, r $= -0.51$) (Figure 3E). The orientation selectivity index did not change after tetanization, indicating that the amplitudes of responses to the optimal and to the orthogonal orientations of the stimuli decreased proportionally (Figure 3G).

For those parameters that changed significantly after tetanization, we separately analyzed the changes among simple and complex cells. There was no significant difference between simple ($n = 6$) and complex ($n = 7$) cells in the decrease in the responses to neither the optimal orientation ($75.4 \pm 10\%$ and $79.6 \pm 3\%$), nor the nonoptimal (orthogonal to the optimal) orientation ($82.7 \pm 12\%$ and $93.7 \pm 15\%$), nor the direction selectivity index ($44.5 \pm 50\%$ and $57.4 \pm 27\%$) (n.s., Student's *t*-test in all three comparisons). This result confirms the validity of pooling together the data for simple and complex cells.

Next, we compared the tuning of cells to the orientation and direction of the stimuli before and after tetanization. For the calculation of averaged orientation tuning, we re-centered the tuning of each cell, so that the orientation of the optimal stimulus was at 0 deg, and then mirrored the tuning plotted in polar coordinates along the optimal orientation. Then, we averaged the responses to stimuli ±30 deg, to stimuli ±60 deg, and so on. Figure 4A,B shows the resulting "half" tuning curves plotted in Cartesian coordinates for the tetanus and no tetanus groups. Such plots represent changes in the whole tuning, rather than just in the responses to optimal and orthogonal orientations, induced by optogenetic tetanization. As described above, after tetanization, there was a significant decrease in

the amplitude of responses to the optimal and orthogonal orientations (Figure 4A). In addition, there was an increase in the amplitude of responses to the stimuli oriented 120 and 150 degrees from the optimal (0°: 78.1 ± 27% $p = 0.0098$, r = −1.02; 90°: 87.9 ± 31% $p = 0.0406$, r = −0.69; 120°: 125 ± 29%, $p = 0.0406$, r = 0.64; 150°: 116 ± 26%, $p = 0.0406$, r = 0.64; Wilcoxon rank test adjusted for multiple comparison using a false discovery rate method). In the control (no tetanus) group, no significant changes in the responses to the stimuli of any orientation were found (Figure 4B). To represent changes in the directional tuning, we redraw the "whole" tuning curve from Figure 4A in polar coordinates (Figure 4C). As a result of the above changes after optogenetic tetanization: a decrease in the responses to the stimuli of optimal and orthogonal orientation, no changes of responses to null orientation, and an increase in responses to 120 and 150 deg, the curve became more round-shaped, i.e., the cells became less selective.

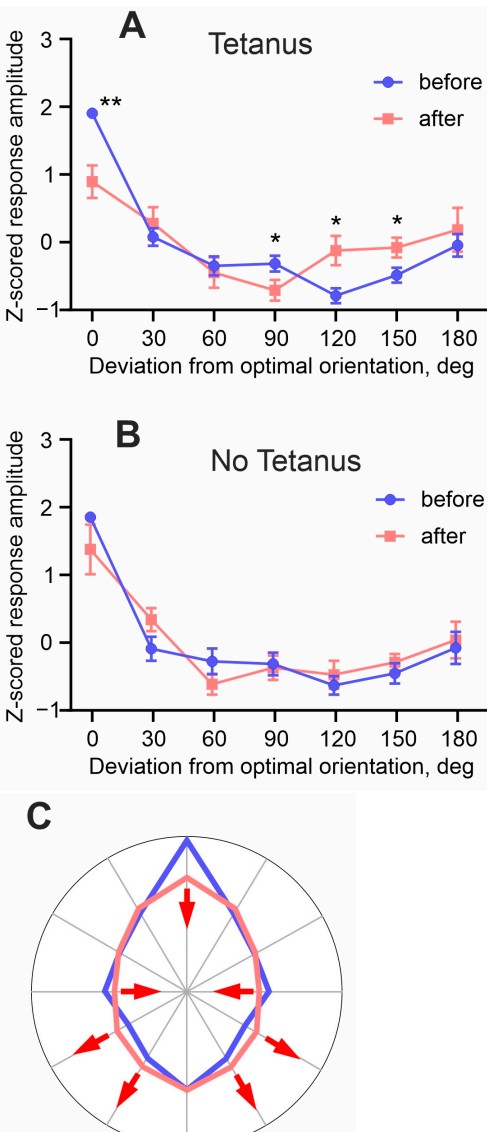

**Figure 4.** Optogenetic tetanization induces changes of the tuning of neurons to orientation and direction of movement of visual stimuli. (**A**,**B**)—Averaged tuning of responses of cortical neurons to orientation before (blue) and after (pink) tetanization (**A**) or before and after a stimulation pause in the control group (**B**). Each point represents the average responses to the grating of one orientation moving in two opposite directions. To facilitate comparison, all values were Z-scored. * $p < 0.05$; ** $p < 0.01$. (**C**)—A schematic representation of changes in the responses to different directions of movement of visual stimuli, induced by optogenetic tetanization of mouse visual cortex neurons.

## 4. Discussion

The results of our study show that high-frequency bursts of action potentials induced in visual cortex neurons using direct optogenetic stimulation (optogenetic tetanization) can cause significant long-term changes in the neurons' responses to visual stimuli. Changes in the responses to visual stimuli reflect changes of synaptic inputs to the neuron and, because optogenetic tetanization is a mostly postsynaptic protocol applied without presynaptic activation, such changes were mediated by heterosynaptic mechanisms. These results provide clear evidence for heterosynaptic plasticity in the whole brain in vivo. Moreover, optogenetic tetanization had distinct effects on responses to stimuli of different orientations, indicating that heterosynaptic plasticity non-uniformly affected neuronal inputs.

### 4.1. Homosynaptic and Heterosynaptic Plasticity—Terminology

The terms homosynaptic and heterosynaptic are used in somewhat different ways in the field of plasticity studies in mammalian preparations and in invertebrate research. Here, we use this terminology in a way that is conventional for plasticity research in mammals. Homosynaptic plasticity is induced at synapses which were presynaptically activated during the induction. It can be induced by protocols such as diverse patterns of afferent tetanization and pairing protocols: stimulation of weak synaptic inputs together with the activation of strong inputs or postsynaptic depolarization [2,3,37]. If activity at both the presynapse and postsynapse is required for the induction, homosynaptc plasticity is associative; a classical example, here, is NMDA-dependent plasticity [3]. If plasticity induction depends only on strong presynaptic activity, but does not require activity (depolarization and spiking) of the postsynaptic cell, homosynaptic plasticity is non-associative, such as at mossy fiber-CA3 synapses in the hippocampus [4]. Heterosynaptic plasticity does not require activity in the presynapse during the induction. In the field of synaptic plasticity in the hippocampus, this term was introduced by Lynch and colleagues [6], who found that tetanization-induced LTP at synapses terminating at apical dendrites of CA1 pyramids is accompanied by LTD at synapses on the basal dendrites, and vice versa; tetanus-induced LTP at the basal dendrites was accompanied by LTD at the apical dendrites. Because synapses undergoing LTD were not stimulated during the induction, this form of LTD was called heterosynaptic. Moreover, heterosynaptic plasticity, both LTP and LTD, could be induced in cortical neurons by episodes of strong postsynaptic activity alone, without stimulation of any synapses during the induction [16,17,24,25,27,38,39]. Local forms of heterosynaptic plasticity include induction of LTP and LTD around the location of the tetanized synapses, forming a Mexican hat-shaped profile of plastic changes [19–21], and spread of plastic changes to non-stimulated synapses of the same neuron, or even to closely located but non-stimulated neurons [22,23]. A distinctive feature of these forms of plasticity is that they are induced at synapses which were not active during the application of the induction protocol and, thus, are heterosynaptic (see [5] for a review).

In invertebrate research, the terms homosynaptic and heterosynaptic are used in a different way. Kandel and Tauc [7,8], studying the facilitation of synaptic transmission in Aplysia, introduced the term heterosynaptic to describe the pairing of a weak stimulation of one pathway with a strong stimulation of another pathway ('heterosynaptic pairing'). Such pairing may lead to a short-lived (9 min on average, up to 40 min in extreme cases) facilitation of responses in the weakly stimulated pathway. Further studies of gill-withdrawal in Aplysia demonstrated the importance of serotonergic modulation during the strong stimulation for both behavioral reaction and heterosynaptic facilitation [40]. Later, these ideas were generalized and modulatory input-dependent plasticity was referred to as heterosynaptic, while Hebbian activity-dependent was referred to as homosynaptic [9]. The term homosynaptic plasticity was also used [9] as a synonym of the property of cooperativity of LTP in mammalian plasticity research [3]. Notably, in the original terminology [7,8], heterosynaptic pairing in Aplysia research roughly corresponds to pairing-induced homosynaptic plasticity and also to the features of cooperativity and associativity of LTP in the hippocampus [3].

### 4.2. Plasticity of Responses and Receptive Fields in Visual Cortex Neurons

Prior research on the plasticity of visual responses and receptive fields of neurons in the visual cortex in vivo was focused on the modifications induced by associative pairing procedures. As we demonstrated in our prior work, pairing the visual stimulus of non-optimal orientation with the optogenetically induced firing of pyramidal neurons in the mouse visual cortex induced long-term changes in the response properties of the stimulated cell, whereby responses to a previously non-optimal (paired) stimulus increased, while responses to the originally preferred orientation of the stimulus decreased [12]. In the cat visual cortex, the pairing of flashing gratings of near-optimal and optimal orientation led to a significant shift in orientation preference [41], and the pairing of the stimulation of non-responsive locations neighboring to the neuron's receptive field with the stimulation of the receptive field center could lead to an increase in the RF toward the stimulated site [10]. Pairing visual stimulation with intracellularly evoked spikes in neurons of the rat visual cortex induced bidirectional modifications of responses. Importantly, responses evoked by the stimulation of unpaired locations could change too, indicative of heterosynaptic changes [11]. Heterosynaptic changes were also reported in a study using imaging of calcium signals in spines of mouse visual cortex neurons [29]. Pairing visual stimulation that activated a spine with postsynaptic spiking induced LTP in the activated spine and heterosynaptic LTD at some of the other spines on the same neuron. With a reservation that the absence of presynaptic activation at the inputs which were not intentionally stimulated is difficult to control with strong visual stimulation and large subthreshold receptive fields, these results indicate that heterosynaptic plasticity can be induced in the visual cortex neurons of rats and mice.

### 4.3. Paradigm of Optogenetic Tetanization to Study Heterosynaptic Plasticity

The paradigm of optogenetic tetanization, implemented in our study, is a transfer to in vivo conditions of the protocol of intracellular tetanization, which is established in brain slice preparations and leads to bi-directional, long-term plastic changes of synaptic inputs [17,25] (see [5] for a review). While the high-frequency bursting of cells induced by pulses of depolarization current or light is an artificial pattern, it clearly corresponds to some patterns of activity in the cortical neurons observed in vivo. Pyramidal neurons in the rat hippocampus can generate bursts of action potentials with frequencies of up to 200 Hz during spontaneous activity [42]. In the cat visual cortex, patterns of high frequency activity comparable to optogenetic tetanization can be produced by optimal visual stimulation [35]. Similar patterns of spiking are also typical for slow-wave sleep, with periodic transitions of cortical neurons between active (up) and silent (down) states [43–45]. An absence of sensory input from the thalamus and the predominantly intracortical origin of activity during slow-wave sleep [46] adds to this similarity. Thus, the pattern of spiking induced by optogenetic tetanization is similar to some naturally occurring patterns of activity in cortical neurons, especially to those during slow-wave sleep, which might, therefore, promote induction of heterosynaptic plasticity.

According to the synaptic homeostasis hypothesis, learning during the awake state drives associative synaptic plasticity and the overall strengthening of synapses, while during sleep, synaptic weights are re-normalized, preparing synapses for further learning. In relation to the operation of visual cortex neurons and our present results, it could be hypothesized that visual stimulation received by the animal when awake leads to adaptive changes in the orientation selectivity of neurons, and high-frequency activity bursts generated during slow-wave sleep, by mechanisms of heterosynaptic plasticity, re-balance neuronal inputs and transfer neurons to some "basic" less-tuned state, thus preparing them for processing new visual information in the next period of wakefulness. An important question for future studies is to reveal how such a re-balancing of synaptic weights interacts with the long-term changes in visual responses induced by tetanization, without compromising lasting synaptic changes which mediate learning to discriminate new stimuli.

### 4.4. Limitations of the Present Study

Using juxtacellular registration, we can only record action potentials generated by the cells, but not the subthreshold changes in membrane potential. However, membrane potential responses to visual stimulation provide additional, more accurate information about the changes in the visual responses and receptive field structure of neurons, as compared to spike responses. Therefore, it is possible that the relatively small effects of optogenetic tetanization reported here were due to the recording of spikes rather than membrane potential changes. To circumvent this limitation, further studies using intracellular recordings from primary visual cortex neurons are necessary. The use of intracellular recordings will also eliminate two further potential confounds of the present study. First, the problem of the standardization of the tetanization protocol, since different levels of oChIEF expression in different neurons in our experiments forced us to adjust the frequency of optogenetic tetanization between 75 and 100 Hz in different cells. Second, intracellular recordings will allow us to confirm that the observed changes in visual responses after intracellular tetanization are resulting from the evoked bursts of action potentials and are not due to other factors arising from the laser radiation entering the brain tissue, such as tissue heating. However, the letter scenario is implausible because previous research showed that the light intensities used in our experiments could cause brain heating by less than 0.5 degrees and, thus, are unlikely to lead to the observed long-term changes in the response properties of neurons [47].

### 4.5. Is Optogenetic Tetanization a Purely Postsynaptic Induction Protocol?

One potential confounding issue of interpreting all the observed changes in visual responses as resulting from heterosynaptic plasticity is the possibility of presynaptic activity at the neuron's inputs during optogenetic tetanization, due to either spontaneous activity or the activation of nearby oChIEF-containing presynaptic neurons or fibers by optogenetic stimulation.

In the absence of visual stimulation during the induction, the probability of repeated presynaptic activation of the neuron's inputs due to spontaneous activity, which could lead to 'spurious STDP' pairing with postsynaptic spikes, is negligibly small. Indeed, plasticity induction requires consistent and repetitive pre-before-post activation for LTP, or post-before-pre for LTD, for >30 times [11,48]. With typical spontaneous firing rates <5 Hz, a presynaptic spike could hit a ~50–100 ms STDP window [2,11,49] of light-induced postsynaptic spikes; however, the probability that such a coincidence will occur >30 times consistently only in the LTP (or only in the LTD) window is extremely low. Rather, presynaptic spikes that occasionally hit the LTP and LTD windows will do so at random, canceling the effects of each other. Therefore, we consider such scenario of 'spurious STDP' unrealistic.

For the following reasons, we consider that the possible presynaptic activation of the neuron's test inputs, due to the activation of nearby oChIEF-containing neurons or fibers during optogenetic tetanization, contributes little to the observed changes in the responses. Our measurements and calculations show that only a small fraction of the light coming out of the optical fiber was reaching the brain (Figure 2A inset; see Methods for detail). We calculated that the light intensity at the tip of the pipette in our experiments was 1.2 mW/mm$^2$. Based on blue light absorption by the brain tissue [50], we estimated that, at a distance of 200 μm from the pipette tip, the light intensity dropped to 0.1 mW/mm$^2$, which is the threshold intensity for inducing action potentials in ChR2-expressing neurons [51]. Therefore, we estimated that the area of brain tissue where the suprathreshold intensity of the light could be reached was approximately a cylinder of 200 μm diameter and 700 μm height when the pipette was inserted to a depth of 500 μm. We further estimate that about 50 L5 neurons and 80 L2/3 neurons expressing oChIEF are located within the volume of possible light activation. Neurons in layer 4 around the recording site are not expressing oChIEF. Given the low probability of monosynaptic connections made by L5 and L2/3 pyramids onto L5 pyramids (about 1/15–1/20) and a small amplitude of excitatory postsynaptic potentials in such connections (typically around 1 mV) [39,52], we assume

that the potential impact of synaptic inputs from neighboring pyramidal neurons activated by light is low. Optical imaging data show that our visual stimuli activate a much bigger cortical area, of a few mm$^2$. Thus, synaptic inputs carrying information about the visual stimulus and arriving at the recording neuron from other pyramids originate from cells in the area that is much bigger than the possible region of supra-threshold light activation. Therefore, even with the potential activation of several synaptic inputs to the recorded cell during optogenetic tetanization, the bulk of inputs mediating responses to visual stimulation originates from neurons outside the light-affected area. Finally, the activation of axon terminals of oChIEF-expressing but distant neurons requires very high light intensity, e.g., for the stimulation of axon terminals of CHR2-containing thalamic neurons, 600 to 9500 mW/mm$^2$ was used [53], which is at least two orders of magnitude higher than the light intensity we used. The possibility of activation of such axons in our experimental conditions was, thus, negligible.

For the above reasons, we conclude that, while the activation of some synaptic inputs to the recorded neurons by optogenetic tetanization cannot be excluded, most of the inputs contributing to the responses to visual stimuli were not activated during optogenetic tetanization and the observed changes of visual responses can be attributed to heterosynaptic plasticity.

*4.6. Possible Mechanisms and Functional Consequences of Plasticity Induced by Optogenetic Tetanization*

Our present results show that after optogenetic tetanization, the most pronounced changes were in the directional selectivity of neurons, and often, the preferred direction of movement was changed to the opposite. This was due to a decrease in the responses to stimuli of the optimal orientation moving in the preferred direction, but no changes in the responses to the opposite (null) direction of movement. Because directional selectivity is created by the interaction of excitatory and inhibitory inputs [54–56], such changes could be mediated by the depression of excitatory inputs or the potentiation of inhibitory inputs, or a combination of both. Indeed, in slices, heterosynaptic plasticity can be induced at both excitatory synapses [15,17,25,57] and at inhibitory GABA-ergic synapses to layer 5 neurons [38]. The present results do not allow us to differentiate between these possibilities. Disentangling the contribution of a decrease in excitation and/or an increase in inhibition requires further studies, e.g., employing intracellular recordings. Our preliminary data from slices of the visual cortex suggest that intracellular tetanization may differentially affect the proximal vs. distal synaptic inputs, with heterosynaptic potentiation dominating at proximal inputs. Whether a similar distance-dependence of heterosynaptic plasticity holds for in vivo conditions, and whether a net potentiation of proximal excitatory and inhibitory synapses induced by intracellular tetanization could account for the observed changes in directional selectivity of visual cortex neurons remains to be investigated in further experiments.

A decrease in responses to optimal and orthogonal orientations, no significant change of responses to the null direction, and an increase in responses to oblique orientations after optogenetic tetanization made the tuning of cells more round-shaped and responses less selective (scheme in Figure 4C). Several aspects of these results are worth noting. First, the decrease in the strong (optimal) responses and the increase in the relatively weak response to oblique orientations have a parallel with the weight-dependence of heterosynaptic plasticity, observed in vivo [11] and extensively studied in slices and computational models [5,58]. Second, the weight-dependence of response changes induced by heterosynaptic plasticity might help to prevent an excessive increase in the responses to optimal stimuli and a runaway decrease in the responses to other-than-optimal stimuli, due to runaway potentiation or depression of the respective inputs due to plasticity governed by Hebbian-type rules. Third, theoretical studies using the information theory approach show that there is an optimal width of tuning of individual neurons for the encoding of three or more features of visual stimuli in population activity [59,60]. Such an optimal width allows

each neuron of a population to encode the maximal amount of information about the visual stimuli. With the tuning width outside the optimal range, the amount of information a neuron can encode is decreasing. We have shown that high-frequency bursts of action potentials evoked by intracellular tetanization lead to a broadening of the neuronal orientational tuning width that might be mediated by mechanisms of heterosynaptic plasticity. Since neurons in the primary visual cortex encode multiple features of visual stimuli, such as orientation, direction and speed of motion, size, etc., we hypothesize that response changes induced by heterosynaptic plasticity may help to keep the width of orientation and direction tunings around such optimal values. Thus, we suggest that heterosynaptic plasticity could play a role not only in stabilizing the orientational and directional tunings of individual neurons, but also in maintaining their width within the range optimized for the high efficiency of encoding of properties of visual stimuli in population activity.

Synaptic plasticity in primary sensory cortical areas may underlie the experience-dependent changes of perception [61,62], and homosynaptic plasticity is one of the basic mechanisms mediating perceptual learning [63]. Since heterosynaptic plasticity induced by bursts of action potentials always accompanies homosynaptic plasticity, it is feasible to suggest that heterosynaptic plasticity may also be involved in the mechanisms of perceptual learning.

To summarize, we show that, in neurons of the mouse visual cortex, optogenetic tetanization can induce changes in the responses to visual stimuli. This provides evidence for heterosynaptic plasticity in the whole brain in vivo. Based on the observed response changes, we hypothesize that the homeostatic role of heterosynaptic plasticity, in addition to its importance in preventing runaway dynamics of synaptic weights and the synaptic drive of individual neurons, extends to the level of neuronal population encoding, by stabilizing the tuning of neuronal responses to features of visual stimuli and keeping the tuning width in the range that is optimal for the encoding of multiple features. Finally, because of the similarities between the patterns of activity during slow-wave sleep and optogenetic tetanization, we suggest that heterosynaptic plasticity may act as one of the mechanisms mediating homeostatic function of slow-wave sleep.

**Author Contributions:** Conceptualization, I.V.S., M.A.V. and A.Y.M.; methodology, I.V.S., A.A.O., M.P.S. and A.A.B.; software, I.V.S.; validation, I.V.S.; formal analysis, I.V.S., A.A.O. and M.A.V.; investigation, I.V.S., A.A.O., M.P.S. and A.A.B.; data curation, I.V.S. and A.A.O.; writing—original draft, I.V.S., M.A.V. and A.Y.M.; writing—review and editing, M.A.V. and A.Y.M.; visualization, I.V.S., M.A.V. and A.Y.M.; supervision, A.Y.M.; project administration, A.Y.M.; funding acquisition, A.Y.M. All authors have read and agreed to the published version of the manuscript.

**Funding:** This study was supported by the Russian Science Foundation (grant #20-15-00398P).

**Institutional Review Board Statement:** All experimental procedures were in accordance with the 2010/63/EU directive for the use of laboratory animals. The research protocol was approved by the Ethics Committee of the Institute of Higher Nervous Activity and Neurophysiology, Russian Academy of Sciences (approved on 15 January 2020; protocol No. 1; project # 20-15-00398).

**Informed Consent Statement:** Not applicable.

**Data Availability Statement:** The dataset is available on request from the authors.

**Acknowledgments:** We are grateful to Stanislav Volgushev for advice on statistical analysis.

**Conflicts of Interest:** The authors declare no conflicts of interest.

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
