# Peer review of "Plasticity of Response Properties of Mouse Visual Cortex Neurons Induced by Optogenetic Tetanization In Vivo"

_cimb, doi:10.3390/cimb46040206_

Round 1

Reviewer 1 Report

Comments and Suggestions for Authors

Smirnov et al reports an experiment aimed at demonstrating that heterosynaptic plasticity can be induced in vivo, using the visual cortex as a model. They combined optogenetic stimulation to juxtacellular recordings to determine whether optogenetically evoked bursts of action potentials in visual cortical cells are capable of inducing long-term changes of responses to visual stimuli.

The issue about the homeostatic role of heterosynaptic plasticity in the dynamic of visual cortical network is topical and very interesting. While the study has the merit to approach a stimulating question, the manuscript, in its current form, has some critical limitations that severely reduce the impact of the reported results. Of particular concerns are some methodological aspects,  the great variability in the data and their interpretation.

- One major concern is about the single cell in vivo activity evoked by optogenetic stimulation. This is supposed to be exclusively determined by a postsynaptic induction protocol. However, the fiber stimulation resolution is not at the single cell level. Therefore, it cannot be excluded an activation of some neighboring cell that can be connected with the recorded neuron.

- Figure 3 seems to be underpowered especially given the variance between cells, probably because of the small sample size. First of all a sample size clarification is needed: it is only specified the n = number of cells, the authors should also indicate the number of mice from which these recordings were obtained.

A more detailed description of the statistic should be included (power analysis) to evaluate the strength of the test used to detect or not a significant difference. For example, in panel 3D no significant difference is detected in NON tetanized group, however, 5/6 cells out of 10 show a clear reduction in response amplitude.

Regarding the difference in optimal orientation changes, the authors report that: “Respectively, averaged values of the change differed dramatically between the tetanus group (80.8 ± 73.9 deg) and control group (30 ± 54.8 deg)”. Given the high SD values, there is probably not a statistically significant difference between the two groups.

Moreover, regarding the variability in cell tuning observed, the authors describe in the results section: “To address a potential confounding effect introduced by initially sharper tuning of such cells in the control no tetanus group, we made the following analysis. For this additional analysis we excluded the cells from the no tetanus group which had DSI or OSI deviating by more than three SD (red dashed lines in Figure 3E,G) from the mean values of these indices in the tetanus group. In the remaining control cells with OSI and DSI similar to those of the tetanized group, neither the directional (N = 5) nor the orientational selectivity index (N = 6) showed significant changes”. Although I understand that the study is sampling from a population of eterogeneous cells regarding their tuning properties, clustering the data as reported does not help in the interpretation given the low sample size ( 2 cells out of 5 have a clear reduction in direction selectivity index).

Minor

- Page 2 line 52 “Tetanus induced long term potentiation" should be abbreviated as LTP

- correct in the manuscript the term “juxtracellular” with the correct form “juxtacellular”.

Author Response

We are grateful to the Reviewer for the positive evaluation of our works and for the comments which helped us to improve the paper. Reviewer’s concerns are addressed in the revised manuscript. Below is a point-by-point reply to the comments.

Rev#1
Smirnov et al reports an experiment aimed at demonstrating that heterosynaptic plasticity can be induced in vivo, using the visual cortex as a model. They combined optogenetic stimulation to juxtacellular recordings to determine whether optogenetically evoked bursts of action potentials in visual cortical cells are capable of inducing long-term changes of responses to visual stimuli.

The issue about the homeostatic role of heterosynaptic plasticity in the dynamic of visual cortical network is topical and very interesting. While the study has the merit to approach a stimulating question, the manuscript, in its current form, has some critical limitations that severely reduce the impact of the reported results. Of particular concerns are some methodological aspects,  the great variability in the data and their interpretation.

- One major concern is about the single cell in vivo activity evoked by optogenetic stimulation. This is supposed to be exclusively determined by a postsynaptic induction protocol. However, the fiber stimulation resolution is not at the single cell level. Therefore, it cannot be excluded an activation of some neighboring cell that can be connected with the recorded neuron.

We have expanded and clarified the section in the Discussion "Is optogenetic tetanization a purely postsynaptic induction protocol?" that is entirely devoted to discussion of this issue. In short, while activation of neighboring neurons during optogenetic stimulation of the recorded cell cannot be ruled out, we explain in detail the reasons why we consider that the contribution of such activation to the observed changes in visual responses might be small or negligible. The number of neighboring pyramidal neurons expressing oChIEF and monosynaptically connected to the recorded cell is small, especially compared to the number of cell in the cortical area activated by visual stimuli, the amplitude of such connections is low, and the light power we used is insufficient to excite the axons of oChIEF-containing neurons. Among other things, the fiber in our experiments was completely outside the brain, and only a small fraction of the light coming out of it enters the brain tissue through the recording glass pipette.

- Figure 3 seems to be underpowered especially given the variance between cells, probably because of the small sample size. First of all a sample size clarification is needed: it is only specified the n = number of cells, the authors should also indicate the number of mice from which these recordings were obtained.

We added the number of animals in which neurons were recorded (n=10 animals in each group).

A more detailed description of the statistic should be included (power analysis) to evaluate the strength of the test used to detect or not a significant difference. For example, in panel 3D no significant difference is detected in NON tetanized group, however, 5/6 cells out of 10 show a clear reduction in response amplitude.

We did not perform a priory power analysis before starting the experiments, and performing post-hoc power analysis, as recent statistical studies have shown, is "logically invalid and practically misleading" (quote from the Abstract of the paper in  Curr Psychol. 2020 June ; 39(3): 870-877. doi:10.1007/s12144-018-0018-1; Abstract). The calculated power is just a monotone function of the p value and carries no additional information. For example, for p=0.012, the calculated power value will always be 0.938, regardless of the applied statistical criterion from the group of Z-statistic methods and the sample size. Nevertheless, we have calculated and reported the effect size parameter r (one of the key indicators in power analysis) for all values analyzed in the paper (we use Wilcoxon rank test).

Regarding the data shown in Figure 3D, the differences in the tetanus group were highly significant (p= 0.0002), while in the control group they were non-significant (p=0.13). We have added all p-values, also for non-significant differences, to the text. 

Regarding the difference in optimal orientation changes, the authors report that: “Respectively, averaged values of the change differed dramatically between the tetanus group (80.8 ± 73.9 deg) and control group (30 ± 54.8 deg)”. Given the high SD values, there is probably not a statistically significant difference between the two groups.

The statistical method we used in this case (Anderson-Darling) does not compare the means of the distributions, but the shape of the distributions themselves, so we agree that in this case it is not appropriate to give the means with SD, so we removed it from the text.

Moreover, regarding the variability in cell tuning observed, the authors describe in the results section: “To address a potential confounding effect introduced by initially sharper tuning of such cells in the control no tetanus group, we made the following analysis. For this additional analysis we excluded the cells from the no tetanus group which had DSI or OSI deviating by more than three SD (red dashed lines in Figure 3E,G) from the mean values of these indices in the tetanus group. In the remaining control cells with OSI and DSI similar to those of the tetanized group, neither the directional (N = 5) nor the orientational selectivity index (N = 6) showed significant changes”. Although I understand that the study is sampling from a population of eterogeneous cells regarding their tuning properties, clustering the data as reported does not help in the interpretation given the low sample size ( 2 cells out of 5 have a clear reduction in direction selectivity index).

We have removed the above analysis with the omission of highly divergent cells from the total sample from the paper.

Minor

- Page 2 line 52 “Tetanus induced long term potentiation" should be abbreviated as LTP

Thank you for the comment, corrected.

- correct in the manuscript the term “juxtracellular” with the correct form “juxtacellular”.

Thank you for the comment, corrected.

Reviewer 2 Report

Comments and Suggestions for Authors

The manuscript "Plasticity of Response Properties of Mouse Visual Cortex Neurons Induced by Optogenetic Tetanization In Vivo" investigates how optogenetic stimulation influences synaptic plasticity in mouse visual cortex neurons. It showcases that inducing high-frequency action potentials without presynaptic activation leads to significant changes in how neurons respond to visual stimuli, providing strong evidence for heterosynaptic plasticity in a live mammalian brain. This study differentiates between homosynaptic and heterosynaptic plasticity, employing a methodology that mirrors natural neuronal activity to examine changes in synaptic inputs and neuronal response dynamics.

The research reveals non-uniform effects of heterosynaptic plasticity on neuronal inputs, indicating a sophisticated mechanism by which synaptic plasticity modulates neuronal functionality, particularly in response to stimuli orientation. The manuscript delves into the underlying mechanisms and potential implications of these findings, suggesting that the observed changes in neuronal selectivity could optimize how visual information is encoded across neuron populations, enhancing perceptual efficiency.

In summary, this work provides vital insights into the adaptive mechanisms of the visual cortex, emphasizing the role of heterosynaptic plasticity in neural processing and cognition. The application of optogenetic techniques offers a promising avenue for further exploration into the neural basis of behavior and learning, highlighting the intricacies of synaptic plasticity and its importance in neural circuit functionality.

To improve the quality of your manuscript, the following enhancements are recommended:

Introduction

1. Consider providing a brief, clear definition of both "homosynaptic" and "heterosynaptic" plasticity at their first mention for readers who may not be familiar with these terms. While the distinction is made, a concise explanation or example could improve understanding.

2. The manuscript references numerous studies to support the discussion on synaptic plasticity forms and their mechanisms. It would be beneficial to briefly summarize the key findings of these studies, especially those cited in lines 34-35 and 46-50, to provide context for your work’s contribution to the field.

3. While the choice of intracellular tetanization and optogenetic stimulation is mentioned, elaborating on why these methods are particularly suited for studying heterosynaptic plasticity in vivo compared to other approaches would strengthen the methodological rationale.

4. The mention of considering terminology related to invertebrate research in the discussion (line 40-41) is intriguing. A brief hint at the potential differences or similarities anticipated could spark interest and highlight the breadth of your study’s relevance.

5. The manuscript touches upon the diverse functional and computational roles of distinct forms of plasticity (lines 42-48). Expanding this section to include specific examples or hypotheses about how heterosynaptic plasticity contributes to these roles would enrich the reader's understanding of its significance.

6. Emphasize more clearly the novel aspects of your study in the context of existing literature. What gaps in our understanding does your work address? How does your approach differ from or improve upon previous studies, especially in relation to in vivo evidence for heterosynaptic plasticity?

7. Briefly discuss any technical challenges or limitations encountered in employing juxtracellular recording and optogenetic stimulation. Acknowledging these can provide a more balanced view and set the stage for future research directions.

9. Consider breaking down complex sentences for clarity and ease of reading. For example, lines 77-91 could be divided into shorter sentences to enhance readability. Ensure consistency in terminology and abbreviations throughout the introduction to aid reader comprehension.

Results

10. The choice of juxtacellular recording and optogenetic stimulation is well justified. However, elaborating on why these specific techniques were chosen over others could provide the reader with a better understanding of their advantages in your experimental setup.

11. The experimental design is clearly laid out, from the transduction of neurons with channelrhodopsin to the recording and analysis of neuronal responses. It might be beneficial to briefly discuss any control measures taken to ensure that the observed effects were specifically due to the optogenetic stimulation and not other variables.

12. While the results mention the classification of neurons into simple and complex types based on the F1/F0 ratio, the statistical analysis supporting the significance of changes observed post-tetanization is not detailed. Including statistical tests used to analyze changes in orientation tuning, response amplitude, and selectivity indices would strengthen the results.

13. The changes in neuronal response properties post-tetanization are intriguing. Expanding on how these changes contribute to our understanding of synaptic plasticity, especially in the context of learning and memory, would provide valuable insight. Additionally, discussing the implications of these findings for the functional and computational roles of the visual cortex could enhance the impact of your work.

14. Comparing your findings with those of previous studies, especially regarding the efficacy of optogenetic stimulation in inducing synaptic plasticity, would contextualize your results within the broader field. This could involve a brief discussion of how your results align with or differ from existing literature.

15. Acknowledging any limitations of your study, such as the specificity of the optogenetic stimulation or the generalizability of the findings to other types of neurons or brain areas, would provide a balanced view. Outlining potential future research directions based on your findings could also be valuable.

16. For clarity, specifying the exact parameters used for optogenetic stimulation (e.g., intensity of light, exact frequency, and duration of pulses) could help in the replication of your study and in understanding its context within the field.

17. Concluding this section with a statement on the significance of your findings for the field of neural plasticity, particularly in vivo studies, would highlight the contribution of your work.

18. The use of specific statistical tests (e.g., Anderson-Darling, Wilcoxon matched-pairs signed rank test) is appropriately mentioned, enhancing the credibility of the findings. It would be beneficial to include a brief rationale for choosing these tests over others to provide readers with insight into the analytical decision-making process.

19. The approach to address potential confounding effects due to initially sharper tuning in the control group is commendable. This demonstrates thoroughness in analysis and strengthens the validity of your conclusions. It might be helpful to further elaborate on the criteria for excluding cells based on deviation in DSI or OSI to ensure transparency and replicability.

20. While significant findings are reported, including mean changes and p-values, incorporating effect sizes where applicable could offer additional insights into the magnitude of the observed effects, enhancing the interpretive value of your results.

21. The description of Figures 3 and 4 suggests that they effectively illustrate the study's findings. Ensuring that these figures are accompanied by clear, concise legends that explain the methodology (e.g., calculation of F1/F0 ratio, classification of cells) and highlight key results will aid reader comprehension. Consider discussing whether the visualization techniques used (e.g., polar coordinates for tuning curves) were selected for their ability to best represent the data's characteristics.

22. The manuscript appropriately acknowledges where changes were not statistically significant, which is important for a balanced discussion. Expanding on why some responses might not have shown significant changes could offer valuable insights, perhaps hypothesizing on biological or experimental factors that could contribute to these outcomes.

23. The separate analysis of changes among simple and complex cells, and finding no significant difference in response decreases post-tetanization, is an important aspect of your results. This could be discussed in the context of what is known about the functional roles of these cell types in visual processing and how your findings contribute to understanding their plasticity.

24. The observation that tetanization led to more rounded tuning curves, indicating decreased selectivity, is intriguing. Further discussion on the potential implications of this finding for the neural coding of visual information and how it might affect visual perception or behavior in the mouse would be insightful.

25. While the analysis is robust, discussing any limitations of your analytical approach and suggesting directions for future research based on your findings could provide a pathway for advancing the field. For instance, exploring the long-term stability of these changes or their occurrence in other areas of the brain could be of interest.

26. Given the technical nature of this analysis, including a summary box or a simplified diagram that encapsulates the main findings and their significance could enhance accessibility for readers less familiar with the statistical or methodological nuances.

Discussion

27. The comparison of homosynaptic and heterosynaptic plasticity mechanisms and terminology across different research domains (mammalian vs. invertebrate studies) is informative and sets a clear stage for your contributions. It would be beneficial to further emphasize how your findings advance our understanding of heterosynaptic plasticity, particularly in the in vivo whole-brain context, which has been less explored compared to in vitro studies.

28. While the results on directional selectivity changes and the evidence for heterosynaptic plasticity in vivo are compelling, highlighting the novel aspects of these findings early in the discussion can help readers appreciate the significance of your work. Specifically, clarifying how these results expand our understanding of neural coding in the visual cortex or revise previous models of synaptic plasticity could be impactful.

29. The discussion around potential mechanisms underlying the observed changes (e.g., the balance of excitatory and inhibitory inputs) is intriguing. Expanding on these hypotheses by discussing how they align with or diverge from known mechanisms of synaptic modification could enrich the discussion. Additionally, proposing specific future experiments to test these hypotheses would be valuable.

30. You've suggested that heterosynaptic plasticity may optimize the encoding efficiency of visual stimuli by adjusting the tuning width of neuronal responses. Further elaborating on the potential functional consequences of these adjustments for visual perception and behavior could provide readers with a more comprehensive understanding of the importance of your findings.

31. The manuscript addresses potential confounds related to presynaptic activity during optogenetic tetanization. Expanding on these considerations to discuss any remaining uncertainties or limitations of the optogenetic tetanization approach would strengthen the manuscript. For instance, considering the potential for broader network effects of optogenetic stimulation could be relevant.

32. The suggestion that heterosynaptic plasticity may play a role during slow-wave sleep is fascinating. Discussing how your findings could influence our understanding of sleep's role in brain function, including memory consolidation and homeostatic regulation, could open up interesting avenues for future research. Additionally, considering the translational potential of these findings, such as implications for neurorehabilitation or the treatment of neurodegenerative diseases, would be compelling.

33. While the main focus of the discussion is on the interpretation of findings, briefly mentioning any statistical methods or analyses that were particularly crucial for supporting your conclusions could be helpful. This is especially relevant if novel or sophisticated statistical techniques were employed.

34. Given the technical nature of the topic, including a summary or a simplified diagram that encapsulates the main findings, hypotheses, and their significance could enhance accessibility for readers not deeply familiar with the field of synaptic plasticity.

Comments on the Quality of English Language

Ensure that the manuscript is thoroughly proofread to correct any typographical errors and to streamline language for clarity and precision. This includes checking for consistency in terminology, especially when discussing complex concepts like heterosynaptic and homosynaptic plasticity.

Author Response

We are grateful to the Reviewer for the positive evaluation of our works and for the comments which helped us to improve the paper. Reviewer’s concerns are addressed in the revised manuscript. Below is a point-by-point reply to the comments.

Rev#2

The manuscript "Plasticity of Response Properties of Mouse Visual Cortex Neurons Induced by Optogenetic Tetanization In Vivo" investigates how optogenetic stimulation influences synaptic plasticity in mouse visual cortex neurons. It showcases that inducing high-frequency action potentials without presynaptic activation leads to significant changes in how neurons respond to visual stimuli, providing strong evidence for heterosynaptic plasticity in a live mammalian brain. This study differentiates between homosynaptic and heterosynaptic plasticity, employing a methodology that mirrors natural neuronal activity to examine changes in synaptic inputs and neuronal response dynamics.

The research reveals non-uniform effects of heterosynaptic plasticity on neuronal inputs, indicating a sophisticated mechanism by which synaptic plasticity modulates neuronal functionality, particularly in response to stimuli orientation. The manuscript delves into the underlying mechanisms and potential implications of these findings, suggesting that the observed changes in neuronal selectivity could optimize how visual information is encoded across neuron populations, enhancing perceptual efficiency.

In summary, this work provides vital insights into the adaptive mechanisms of the visual cortex, emphasizing the role of heterosynaptic plasticity in neural processing and cognition. The application of optogenetic techniques offers a promising avenue for further exploration into the neural basis of behavior and learning, highlighting the intricacies of synaptic plasticity and its importance in neural circuit functionality.

To improve the quality of your manuscript, the following enhancements are recommended:

Introduction

  1. Consider providing a brief, clear definition of both "homosynaptic" and "heterosynaptic" plasticity at their first mention for readers who may not be familiar with these terms. While the distinction is made, a concise explanation or example could improve understanding.

We have expanded the definition of types of plasticity at the beginning of the Introduction, which now reads as follows:

Synaptic plasticity can be segregated in two broad groups, homosynaptic and heterosynaptic, defined by the requirement of presynaptic activity at a synapse during the induction. Homosynaptic plasticity requires presynaptic activity at the synapse during the induction, and thus occurs at synapses that were directly involved in the activation of a cell during the induction. Homosynaptic plasticity can be associative, such as canonical NMDA-dependent plasticity in CA1 area of the hippocampus [3], or non-associative, such as plasticity at mossy fiber-CA3 synapses [4]. Heterosynaptic  plasticity does not require presynaptic activation of the synapse during the induction, and thus can  be occur at synapses that were not active during the induction [5]. 

  1. The manuscript references numerous studies to support the discussion on synaptic plasticity forms and their mechanisms. It would be beneficial to briefly summarize the key findings of these studies, especially those cited in lines 34-35 and 46-50, to provide context for your work’s contribution to the field.

We have added following text to the Introduction to address this issue:

This type of heterosynaptic plasticity requires activation of L-type voltage-dependent calcium channels but does not depend on activation of glutamate NMDA receptors [16], which act as molecular detectors of the coincidence of pre- and postsynaptic activity and are critical for canonical homosynaptic Hebbian plasticity [18].   

  1. While the choice of intracellular tetanization and optogenetic stimulation is mentioned, elaborating on why these methods are particularly suited for studying heterosynaptic plasticity in vivo compared to other approaches would strengthen the methodological rationale.

We expanded explanation of using intracellular tetanization in our study in the following way:

Intracellular tetanization paradigm has been applied to study plasticity in diverse cells and preparations: pyramidal neurons in the hippocampus [16], granular neurons of the dentate gyrus [22], excitatory and inhibitory neurons of the neocortex [23–28], and even identified neurons of the common snail [29,30], but has never been tested in in vivo preparations.

  1. The mention of considering terminology related to invertebrate research in the discussion (line 40-41) is intriguing. A brief hint at the potential differences or similarities anticipated could spark interest and highlight the breadth of your study’s relevance.

We have added following text in to Introduction:

In short, the term heterosynaptic plasticity in invertebrate research embraces a broader range of phenomena, including associative changes, non-associative changes and modulatory plasticity.   

  1. The manuscript touches upon the diverse functional and computational roles of distinct forms of plasticity (lines 42-48). Expanding this section to include specific examples or hypotheses about how heterosynaptic plasticity contributes to these roles would enrich the reader's understanding of its significance.

We have added following text in to Introduction:

However, Hebbian-type rules introduce positive feedback on changes of synaptic weights and activity. Indeed, theoretical studies and computer simulations have shown that model neural networks built using only Hebbian-type plastic synapses are intrinsically unstable, with synaptic weights tending to reach extreme saturated values, and activity prone to runaway dynamics, either unrestrained increase or silencing. To counteract these undesired effects, theoretical and simulation studies suggested heterosynaptic plasticity as is a necessary component of learning networks. Indeed, incorporating heterosynaptic plasticiy in the models helps to robustly prevent runaway dynamics, enhance synaptic competition, and increase the contrast of synaptic weight changes [5,13–15].  

  1. Emphasize more clearly the novel aspects of your study in the context of existing literature. What gaps in our understanding does your work address? How does your approach differ from or improve upon previous studies, especially in relation to in vivo evidence for heterosynaptic plasticity?

We have added following text in to Introduction:

Thus, for the first time, we used the protocol of intracellular tetanization, which is well-established in vitro, to study the possible role of heterosynaptic plasticity in the functioning of neuronal networks in vivo.

Here and below are groups of comments from reviewers that are similar in meaning.

  1. Briefly discuss any technical challenges or limitations encountered in employing juxtracellular recording and optogenetic stimulation. Acknowledging these can provide a more balanced view and set the stage for future research directions.
  2. Acknowledging any limitations of your study, such as the specificity of the optogenetic stimulation or the generalizability of the findings to other types of neurons or brain areas, would provide a balanced view. Outlining potential future research directions based on your findings could also be valuable.

We have added “Limitations of the present study” section to Discussion:

Using juxtacellular registration, we can only record action potentials generated by the cells, but not the subthreshold changes in membrane potential. However, membrane potential responses to visual stimulation provide additional, more accurate information about changes of visual responses and receptive field structure of neurons as compared to spike responses. Therefore, it is possible that the relatively small effects of optogenetic tetanization reported here were due to the recording of spikes rather than membrane potential changes. To circumvent this limitation, further studies using intracellular recordings from primary visual cortex neurons are necessary. The use of intracellular recording will also eliminate two further potential confounds of the present study. First, a problem of standardization of the tetanization protocol, since different levels of oChIEF expression in different neurons in our experiments forced us to adjust the frequency of optogenetic tetanization between 75 to 100 Hz in different cells. Second, intracellular recordings will allow to confirm that the observed changes of visual responses after intracellular tetanization are resulting from the evoked bursts of action potentials are not due to other factors arising from the laser radiation entering the brain tissue, such as tissue heating. However, that letter scenario is implausible, because previous research showed that the light intensities used in our experiments could cause brain heating b by less than 0.5 degrees and thus are unlikely to lead to the observed long-term changes in the response properties of neurons [47].

  1. Consider breaking down complex sentences for clarity and ease of reading. For example, lines 77-91 could be divided into shorter sentences to enhance readability. Ensure consistency in terminology and abbreviations throughout the introduction to aid reader comprehension.

Complex sentence has been split into two:

A drawback of using visual or electric stimulation to evoke strong postsynaptic firing necessary for plasticity induction in visual cortex neurons is that such stimuli are associated with activation of a large set of presynaptic fibers that may include the test inputs. This would hinder clear interpretation of plastic changes at test inputs as heterosynaptic. To circumvent this limitation, we used a postsynaptic induction protocol of intracellular tetanization — bursts of postsynaptic spikes evoked using optogenetic stimulation of the recorded neuron without presynaptic activation. Thus, for the first time, we used the protocol of intracellular tetanization, which is well-established in vitro, to study the possible role of heterosynaptic plasticity in the functioning of neuronal networks in vivo.

Results

  1. The choice of juxtacellular recording and optogenetic stimulation is well justified. However, elaborating on why these specific techniques were chosen over others could provide the reader with a better understanding of their advantages in your experimental setup.

We have added in the Results section the following text, explaining the choice of optogenetic stimulation:

However, a pilot series of experiments showed that passing through a juxtacellular electrode current pulses of high amplitude, necessary for the induction of high-frequency bursts of action potentials, resulted in membrane rupture. To circumvent this problem, we decided to use an optogenetic approach for tetanization of neurons. Classical channelrhodopsin2, due to the pronounced sensitization of responses during rhythmic stimulation, allows to induce controlled bursts of action potentials at frequencies up to 20–30 Hz, but not higher [31].

  1. The experimental design is clearly laid out, from the transduction of neurons with channelrhodopsin to the recording and analysis of neuronal responses. It might be beneficial to briefly discuss any control measures taken to ensure that the observed effects were specifically due to the optogenetic stimulation and not other variables.

We have added the following text to the Discussion section to address this issue:

Second, intracellular recordings will allow to confirm that the observed changes of visual responses after intracellular tetanization are resulting from the evoked bursts of action potentials are not due to other factors arising from the laser radiation entering the brain tissue, such as tissue heating. However, that letter scenario is implausible, because previous research showed that the light intensities used in our experiments could cause brain heating b by less than 0.5 degrees and thus are unlikely to lead to the observed long-term changes in the response properties of neurons [47].

  1. While the results mention the classification of neurons into simple and complex types based on the F1/F0 ratio, the statistical analysis supporting the significance of changes observed post-tetanization is not detailed. Including statistical tests used to analyze changes in orientation tuning, response amplitude, and selectivity indices would strengthen the results.

Since there were no significant changes in responses to the optimal and non-optimal orientation and directional selectivity index after tetanization between simple and complex cell (divided by F1/F0 ratio) we have analyzed them together. There is a following text addressing this issue:

For those parameters that changed significantly after tetanization, we analyzed separately the changes among simple and complex cells.  There was no significant difference between simple (n=6) and complex (n=7) cells in the decrease of the responses to neither the optimal orientation (75.4±10% and 79.6±3%), nor the nonoptimal (orthogonal to the optimal) orientation (82.7±12% and 93.7±15%), nor the direction selectivity index (44.5±50% and 57.4±27%) (n.s., Student's t-test in all three comparisons).  This result confirms the validity of pooling together the data for simple and complex cells.

  1. The changes in neuronal response properties post-tetanization are intriguing. Expanding on how these changes contribute to our understanding of synaptic plasticity, especially in the context of learning and memory, would provide valuable insight. Additionally, discussing the implications of these findings for the functional and computational roles of the visual cortex could enhance the impact of your work.

We have expanded the discussion of the possible role of our findings in the functional and computational features of the visual cortex (in sections of the Discussion "Paradigm of optogenetic tetanization..."   and "Possible mechanisms and functional consequencies...". We added several testable hypotheses and directions of further research.  In brief, we suggest that heterosynaptic plasticity could play a role not only in stabilizing the orientational and directional tunings of individual neurons, but also in maintaining their width within the range optimized for high efficiency of encoding of properties of visual stimuli in population activity; and play a role in synaptic homeostasis over the sleep-wake cycle.

Addittionaly, we have added new text on possible role of heterosynaptic plasticity in perceptual learning:  

Synaptic plasticity in primary sensory cortical areas may underlie experience-dependent changes of perception [58] [59], and homosynaptic plasticity is one of the basic mechanisms mediating perceptual learning [60].  Since heterosynaptic plasticity induced by bursts of action potentials always accompanies homosynaptic plasticity, it is feasible to suggest that heterosynaptic plasticity may also be involved in the mechanisms of perceptual learning.

  1. Comparing your findings with those of previous studies, especially regarding the efficacy of optogenetic stimulation in inducing synaptic plasticity, would contextualize your results within the broader field. This could involve a brief discussion of how your results align with or differ from existing literature.

We have added the following text in to the Results section:

Previously, it has been shown that high-frequency light stimulation can reliably induce classical LTP in oChIEF-expressing neurons [31].

  1. For clarity, specifying the exact parameters used for optogenetic stimulation (e.g., intensity of light, exact frequency, and duration of pulses) could help in the replication of your study and in understanding its context within the field.

There is following text addressing the issue in the Methods section:

Each train consisted of 8-10 pulses with the frequency of 75-100 Hz and pulse duration of 8 ms. The frequency and the number of pulses were adjusted individually for each cell in order to induce APs with the target frequency of 100 Hz or maximal possible frequency for that cell. 

Light intensity, measured at the tip of the optical fiber with the maximal LED power, was around 5 mW (power meter PM16-130, ThorLabs).

  1. Concluding this section with a statement on the significance of your findings for the field of neural plasticity, particularly in vivo studies, would highlight the contribution of your work.

  2. The comparison of homosynaptic and heterosynaptic plasticity mechanisms and terminology across different research domains (mammalian vs. invertebrate studies) is informative and sets a clear stage for your contributions. It would be beneficial to further emphasize how your findings advance our understanding of heterosynaptic plasticity, particularly in the in vivo whole-brain context, which has been less explored compared to in vitro studies.

  3. While the results on directional selectivity changes and the evidence for heterosynaptic plasticity in vivo are compelling, highlighting the novel aspects of these findings early in the discussion can help readers appreciate the significance of your work. Specifically, clarifying how these results expand our understanding of neural coding in the visual cortex or revise previous models of synaptic plasticity could be impactful.

Significance of our findings is described in the last paragraph of Discussion:

To summarize, we show that, in neurons of mouse visual cortex, optogenetic tetanization can induce changes of responses to visual stimuli. This provides evidence for heterosynaptic plasticity in the whole brain in vivo. Based on the observed response changes we hypothesize that homeostatic role of heterosynaptic plasticity, in addition to its importance in preventing runaway dynamics of synaptic weights and synaptic drive of individual neurons, extends to the level of neuronal population encoding, by stabilizing the tuning of neuronal responses to features of visual stimuli and keeping the tuning width in the range that is optimal for encoding of multiple features. Finally, because of the similarities between the patterns of activity during slow-wave sleep and optogentic tetanization, we suggest that heterosynaptic plasticity may act as one of the mechanisms mediating homeostatic function of the slow wave sleep.

  1. The use of specific statistical tests (e.g., Anderson-Darling, Wilcoxon matched-pairs signed rank test) is appropriately mentioned, enhancing the credibility of the findings. It would be beneficial to include a brief rationale for choosing these tests over others to provide readers with insight into the analytical decision-making process.

We have added the following text in to the Methods section:

For experimental and control groups in which distribution of data did not pass the normality test, we used nonparametric tests (e.g. Wilcoxon test), and for data which passed normality test parametric tests (e.g. t-test) were used. Specific tests are indicated in the results.

  1. The approach to address potential confounding effects due to initially sharper tuning in the control group is commendable. This demonstrates thoroughness in analysis and strengthens the validity of your conclusions. It might be helpful to further elaborate on the criteria for excluding cells based on deviation in DSI or OSI to ensure transparency and replicability.

We did not exclude cells from the analysis based on their DSI or OSI.

  1. While significant findings are reported, including mean changes and p-values, incorporating effect sizes where applicable could offer additional insights into the magnitude of the observed effects, enhancing the interpretive value of your results.

We included the effect size wherever the samples were compared and a p-value was given.

  1. The description of Figures 3 and 4 suggests that they effectively illustrate the study's findings. Ensuring that these figures are accompanied by clear, concise legends that explain the methodology (e.g., calculation of F1/F0 ratio, classification of cells) and highlight key results will aid reader comprehension. Consider discussing whether the visualization techniques used (e.g., polar coordinates for tuning curves) were selected for their ability to best represent the data's characteristics.

We have added following text describing Figure 4:

Such plots represent changes of the whole tuning, rather than just of responses to optimal and orthogonal orientations, induced by optogenetic tetanization.

  1. The manuscript appropriately acknowledges where changes were not statistically significant, which is important for a balanced discussion. Expanding on why some responses might not have shown significant changes could offer valuable insights, perhaps hypothesizing on biological or experimental factors that could contribute to these outcomes.

We expanded text describing possible biological role of significant and non-significant changes found:

A decrease of responses to optimal and orthogonal orientations, no significant change of responses to null direction and an increase of responses to oblique orientations after optogenetic tetanization made the tuning of cells more round-shaped, and responses less selective (scheme in Figure 4C).

  1. The separate analysis of changes among simple and complex cells, and finding no significant difference in response decreases post-tetanization, is an important aspect of your results. This could be discussed in the context of what is known about the functional roles of these cell types in visual processing and how your findings contribute to understanding their plasticity.

In fact, we did not find any significant differences between changes in responses of simple and complex cells after optogenetic tetanization; therefore, we don't discuss their different roles in visual processing.

  1. The observation that tetanization led to more rounded tuning curves, indicating decreased selectivity, is intriguing. Further discussion on the potential implications of this finding for the neural coding of visual information and how it might affect visual perception or behavior in the mouse would be insightful.

  2. The suggestion that heterosynaptic plasticity may play a role during slow-wave sleep is fascinating. Discussing how your findings could influence our understanding of sleep's role in brain function, including memory consolidation and homeostatic regulation, could open up interesting avenues for future research. Additionally, considering the translational potential of these findings, such as implications for neurorehabilitation or the treatment of neurodegenerative diseases, would be compelling.

We have added the following text to the discussion:

In relation to operation of visual cortex neurons and our present results, it could be hypothesized that visual stimulation received by the animal when awake leads to adaptive changes in the orientation selectivity of neurons, and high-frequency activity bursts generated during slow-wave sleep, by mechanisms of heterosynaptic plasticity, re-balance neuronal inputs and transfer neurons to some "basic" less tuned state, thus preparing them for processing new visual information in the next period of wakefulness. An important question for future studies is to reveal how such re-balancing of synaptic weights interacts with long-term changes in visual responses induced by tetanization, without compromising lasting synaptic changes which mediate learning to discriminate new stimuli.

  1. While the analysis is robust, discussing any limitations of your analytical approach and suggesting directions for future research based on your findings could provide a pathway for advancing the field. For instance, exploring the long-term stability of these changes or their occurrence in other areas of the brain could be of interest.

We have added the following text to the discussion:

An important question for future studies is to reveal how such re-balancing of synaptic weights interacts with long-term changes in visual responses induced by tetanization, without compromising lasting synaptic changes which mediate learning to discriminate new stimuli.

  1. Given the technical nature of this analysis, including a summary box or a simplified diagram that encapsulates the main findings and their significance could enhance accessibility for readers less familiar with the statistical or methodological nuances.

  2. Given the technical nature of the topic, including a summary or a simplified diagram that encapsulates the main findings, hypotheses, and their significance could enhance accessibility for readers not deeply familiar with the field of synaptic plasticity.

We have a graphical abstract visualizing the methodology and main findings of our study, which unfortunately was not included in the last submission for some reasons. We hope that the graphical abstract will be included in the final version of the paper.

Discussion 

  1. The discussion around potential mechanisms underlying the observed changes (e.g., the balance of excitatory and inhibitory inputs) is intriguing. Expanding on these hypotheses by discussing how they align with or diverge from known mechanisms of synaptic modification could enrich the discussion. Additionally, proposing specific future experiments to test these hypotheses would be valuable.

We have added the following text to the discussion:

Our preliminary data from slices of visual cortex suggest that intracellular tetanization may differentially affect the proximal vs. distal synaptic inputs, with heterosynaptic potentiation dominating at proximal inputs. Whether similar distance-dependence of heterosynaptic plasticity holds for in vivo conditions, and whether a net potentiation of proximal excitatory and inhibitory synapses induced by intracellular tetanization could account for the observed changes in directional selectivity of visual cortex neurons remains to be investigated in further experiments.

  1. You've suggested that heterosynaptic plasticity may optimize the encoding efficiency of visual stimuli by adjusting the tuning width of neuronal responses. Further elaborating on the potential functional consequences of these adjustments for visual perception and behavior could provide readers with a more comprehensive understanding of the importance of your findings.

We have added the following text to this section:

We have shown that high-frequency bursts of action potentials evoked by intracellular tetanization lead to a broadening of the neuronal orientational tuning width that might be mediated by mechanisms of heterosynaptic plasticity. 

  1. The manuscript addresses potential confounds related to presynaptic activity during optogenetic tetanization. Expanding on these considerations to discuss any remaining uncertainties or limitations of the optogenetic tetanization approach would strengthen the manuscript. For instance, considering the potential for broader network effects of optogenetic stimulation could be relevant.

There is a section in the Discussion titled "Is optogenetic tetanization a purely postsynaptic induction protocol?" that is entirely devoted to discussing this issue. In the revised version of the manuscript, we have expanded and clarified this section. In short, while it is true that activation of neighboring neurons during optogenetic stimulation of the recorded cell cannot be completely ruled out, we explain in detail why it is likely that this contribution is negligible or not involved in the recorded changes in visual responses. The number of neighboring pyramidal neurons expressing oChieff and monosynaptically connected to the recorded cell is small, the amplitude of these connections is very low, and the light power we used is insufficient to excite the axons of oChieff-containing neurons.

  1. While the main focus of the discussion is on the interpretation of findings, briefly mentioning any statistical methods or analyses that were particularly crucial for supporting your conclusions could be helpful. This is especially relevant if novel or sophisticated statistical techniques were employed.

We emphasized several times in the discussion text that our results were statistically significant. 

Round 2

Reviewer 2 Report

Comments and Suggestions for Authors

The authors have thoroughly addressed all of my previously raised concerns, and with these revisions, I believe the manuscript is now in a state suitable for acceptance. I commend the authors for their diligent efforts in refining the manuscript and recommend its publication in its current form.

Comments on the Quality of English Language

Minor editing of English language required